# A General Overview of Overhead Multi-Station Multi-Shuttle Systems and the Innovative Applications Trend in Vietnam



**Thuy Duy Truong** [1,2,3], **Nguyen Huu Loc Khuu** [1,2,3], **Quoc Dien Le** [1,2,3], **Tran Thanh Cong Vu** [2,3], **Hoa Binh Tran** [2,3] and **Tuong Quan Vo** [1,2,3,*]

1   Department of Mechatronics, Faculty of Mechanical Engineering, Ho Chi Minh City University of Technology (HCMUT), 268 Ly Thuong Kiet Street, District 10, Ho Chi Minh City, Vietnam; ttduy.sdh221@hcmut.edu.vn (T.D.T.)
2   Bach Khoa Research Center for Manufacturing Engineering, Ho Chi Minh City University of Technology (HCMUT), 268 Ly Thuong Kiet Street, District 10, Ho Chi Minh City, Vietnam
3   Vietnam National University Ho Chi Minh City, Linh Trung Ward, Thu Duc City, Ho Chi Minh City, Vietnam
*   Correspondence: vtquan@hcmut.edu.vn; Tel.: +84-933-32-7078

**Abstract:** Research and development on a global scale have been conducted on overhead hoist transportation systems (OHTSs) in recent years. The majority of these systems are utilized in manufacturing facilities that are either semiautomated or fully automated. By using stochastic models to evaluate medication distribution and product delivery processes in automated delivery systems, hospitals can reduce patient waiting times and drug response times. Warehouses are being transformed into fully automated fulfillment factories by using conveyors and shelf-lifting mobile robots, which reduce waiting times and improve efficiency. Modern warehouses are increasingly becoming fully automated fulfillment facilities as a response to the significant development of e-commerce. A significant number of organizations are using mobile robots or conveyor systems to transport shelves. The parts-to-picker model is used to transport stock-keeping units (SKUs) to stationary pickers at picking workstations. The aim of this study is to analyze and organize the relationship between transportation system families. They are utilized in various fields, such as warehouses, hospitals, airports, cross-dockings, etc. Furthermore, this study categorizes a range of synchronization issues that arise from minor variations in workstation configurations within different warehouse settings. Next, we identify a multistation ATS (automatic transportation system) that switches lines to different stations by using overhead conveyors and active line-switching devices. Vietnam's automated freight problem can be solved with this potential solution. Our study's findings suggest that enhancing the workstation layout can significantly enhance throughput performance. As a result, the benefits of synchronization can surpass those provided by other well-studied decision tasks.

**Keywords:** literature review; overhead; transportation; delivering; station; shuttle; routes





## 1. Introduction

The development of warehouse automation has been rapid since the turn of the century. This advancement has been greatly aided by the use of autonomous vehicle-based storage and retrieval (AVS/R) or shuttle-based storage and retrieval (SSR) systems. Shelves with aisles are utilized in these systems and autonomous transports are deployed on every level in each aisle. Vertical conveyance is possible through the use of lifts. Another significant advancement is the automation of pallet stacking and destacking, as well as the development of mixed-case palletizing technology in the early 2000s. A new stage of automated guided vehicles (AGVs) that aid in the order picking process has been introduced recently. Eventually, these systems will result in automated selection procedures [1]. The well-being of the workplace depends on the ergonomics of operators.

Bey-Temsamani et al. proposed that an intelligent overhead hoist transportation system is to enhance an operator's ergonomics and productivity, with semiautomatic movement, wireless controls, labeling and LED-indicators, and an intelligent gripping system [2]. It is necessary for them to meet multiple strict industrial standards, which include physical load, speed and place control, function specification, effectiveness, and instructions. The ancient human desire to transport large loads led to the invention of wheelbarrows and outer shells [3].

This enhances productivity, reduces cycle time, and safeguards against injury. This requirement is being heightened due to rapidly decreasing production sizes and demands for mass customization. However, the conventional (hoist-based) devices such as jacks, cranes, and rope hoists are currently available for industrial hoist devices. The inertia compensation is inadequate or nonexistent, making it difficult to use and exert dynamic stresses on the worker's body [4]. Operators may refuse to use established systems in certain circumstances, which can have negative physical effects and long-term health concerns. This impact is most noticeable for lighter weights because operators find it difficult to resist using assisting systems for larger loads (i.e., greater than 100 kg) [2].

On the other hand, human transportation in hospitals typically provides patients with handcarts when delivering conventional medications from hospital stores to patients. The prolonged process of administering medication decreases its effectiveness and increases the risk of infection. A recent study of registered nurses revealed that time was spent on tasks that were not related to patient care, like filling up on supplies. The main factor that impacted nurses' time at the bedside [5] was this. The standard of care can be improved by removing caregivers from tasks that are not related to patient care. To improve these problems, collaborative robotics were applied in multiple aspects of manufacturing and hospitals. Despite the fact that commercial collaborative robotics are already secure and use smart control principles, they are unable to handle the volume of payloads required by the industry.

Both academics and practitioners acknowledge the significance of potential business and hospital optimization [6]. The objective of this study is to solve the challenges of synchronization in part-to-picker systems in warehouses. In addition, this research also aims to improve throughput efficiency by synchronizing the dispatch process of stock-keeping units with client order processing. The significance of parts-to-picker systems is discussed, the picking process is detailed, and synchronization problems are explained. To characterize the scope of our review and the design of our research trend in this section, especially in Vietnam, we expand on the relevance of parts-to-picker systems. The picking process in parts-to-picker systems will be clarified and our knowledge of synchronization challenges will be defined in the sections that follow.

Warehouses are crucial in supply chains as they buffer material flow, consolidate products, and handle value-added processes. Continuous improvement in production networks is necessary for market competition, which requires higher performance from warehouses. New management philosophies such as just-in-time (JIT) and lean production pose challenges, including tighter inventory control and shorter response times. By implementing information technologies like bar coding, RF, and WMS, warehouse operations can be improved, with real-time control, easy communication, and high levels of automation [7].

The increase in e-commerce sales has led to a shift from conventional picker-to-parts arrangements to mission-critical fulfillment factories, making traditional picker-to-parts configurations inefficient [8]. Human order pickers frequently waste time by walking or driving. By providing stock-keeping units (SKUs) to stationary pickers, parts-to-picker systems enable workers to concentrate on obtaining items requested by picking orders. Shelf-lifting mobile robots or an automated storage and retrieval system (ASRS) are used to supply stationary pickers. The storage buffer allows for a significant separation of local warehouse operations from other activities taking place elsewhere in the distribution network. Therefore, most warehousing literature focuses on specific warehouse issues (e.g., [9–11]).

Warehouse operations are labor-intensive and require extensive locations for amenities such as racking, stock, unloading, and docking vehicles. Businesses can now store millions of unique goods and handle massive and fluctuating daily order volumes due to the development of e-commerce. Order picking is the most time-consuming and expensive procedure, it is repetitive, frequently suffers from poor ergonomics, and requires high-quality workers ready to work in shifts. This will result in the development of warehouse automation. This has been considered since the 1960s when the first high-bay unit-load warehouses with aisle-captive cranes driving on tracks were created in Germany.

Automated storage and retrieval (AS/R) systems can store bulk items on unit loads (pallets or totes (mini load system)). Human pick stations are combined to form a parts-to-picker system. In Western Europe alone, there are around 40 completely automated warehouses in service, with several more being under construction [8]. These warehouses are ostensibly more cost-effective and substantially smaller than their traditional, manual equivalents. There are several partially robotized warehouses that have been created in addition to fully automated warehouses. In 2012 and 2016, robot technology was used to complete 63 large new warehouses in the Netherlands [8]. The majority of warehouse research is still focused on traditional storage and order picking approaches. The review by De Koster, Le-Duc, and Roodbergen presented several research possibilities towards semiautomated picking approaches [11].

Automation has made significant progress in material handling in the past few years, and we are not far away from a completely automated fulfillment solution [8]. Parts-to-picker systems are used in various businesses, including e-commerce shops and distribution facilities that provide physical storefronts. SKU bins are used by these systems to transport SKUs to the picker, where they are then placed in client bins. The picker retrieves the required items from the SKU bin, places them in the customer bins, and confirms the choices. In some workstations, bins can be weighed automatically to identify picking faults. The current SKU bin is replaced with the next one, and the selection process is repeated until a customer bin has all the required parts. Once the client bin has been constructed, it is transferred to the packaging and shipping department. Manual picking workstations are capable of handling up to 1000 order lines per hour [11,12].

Logistics service quality (LSQ) is crucial for both logistics service providers (LSPs) and customers, as it ensures customer satisfaction and market position. However, there is still uncertainty about what constitutes LSQ and the best ways to analyze, measure, and improve it. Despite the growing number of papers in this field, there is a lack of comprehensive approaches and models. Through a systematic literature review approach, Kilibarda aimed to provide concrete scientific and practical contributions [13]. Also, this research created new scientific approaches and models for measuring and improving LSQ in different transport and logistics systems. This research literature employed a range of methodologies to assess logistics service quality, which encompassed established frameworks. These methods include SERVQUAL, SERVPERF, LSQ, and the Kano model, as well as statistical approaches and adhering to industry standards. Nevertheless, it is essential to acknowledge that the SERVQUAL model does possess certain limitations, including its lack of universal applicability and its emphasis on reliability as a primary dimension. The review of logistics systems and relevant literature revealed a diminished level of quality in logistics service provision. The investigation of factors contributing to poor quality, including technology, information systems, organizational structure, geographical location, customer relationship management, and human factors, is imperative for the purpose of improvement.

However, choosing remains a difficulty, particularly for huge assortments that include irregular or broken items. Modern fulfillment systems use manned picking workstations to combine SKU bins and client bins in a small space. A human order picker will be able to assemble orders without much travel with this. The picker is able to reach all bins with their arms and only takes a few steps to the left and right. Boysen, Nils, Stefan Schwerdfeger, and Konrad Stephan presented an example which optimizes the problem

for synchronization in an ergonomic picking workstation with a capacity for a single SKU bin at a time and two parallel customer bins [14].

The problem consists of four orders with demands for SKUs A, B, and C. The solution is to consider whether an order demands the SKU but not the requested number of pieces. The picking workstation is equipped with a fully automated injection mechanism for finished customer bins, which can still receive the current SKU. There are two alternative solutions presented: solution (a) requires three SKU bin visits to finish all four orders, while solution (b) demands the delivery of four SKU bins.

By reducing the number of SKU bins needed to fulfill a given set of customer orders, there are two advantages: a reduction in preparation time and alleviation of bin supply. Once pick lists are given, the time it takes for a picker to extract the required items from an SKU receptacle and place them in the appropriate customer bins is set. Each SKU bin exchange necessitates a setup time comprised of a toggling delay for the bin exchange, a picker orientation delay, and a potential source of delayed bin delivery. This is called a transportation delay. Average bin exchanges per order is a common metric in many real-world parts-to-picker warehouses, and this KPI has a strong correlation with maximizing throughput performance. The bin supply system is less burdened by reducing the number of SKU receptacles that need to be delivered to a stationary picker. By doing this, the fleet of mobile robots can be reduced and unanticipated delays can be avoided.

The purpose of addressing synchronization issues in parts-to-picker systems is to enhance performance by achieving synchronization between the SKUs including the SKU bins and the picking orders which accumulated in customer bins. This research aims to categorize the various synchronization challenges encountered in parts-to-picker systems and highlight their significance. The provided framework offers a systematic categorization system and a quantification of the operational advantages associated with various configurations of workstations. Next, we determine the path for future study, which is the use of remedies in industries and medical facilities in Vietnam.

Our survey is not focused on technical aspects, such as the specific hardware used, but instead takes on an operations research perspective. Therefore, our primary focus is on the key decision problems that need to be addressed when planning a prospective automated storage/retrieval system (ASRS) and addressing the storage location assignment problem in warehouse transportation. Given the large amount of ASS applications, it is not surprising that related survey papers already exist. Additionally, we consider the conventional storage and order picking methods, both when modifying an existing system and during its operational stage. Furthermore, the main goal of their survey is to develop analytical techniques that have been specially developed to address layout considerations. Therefore, it is reasonable to argue for a comprehensive survey study on the topic of multistation multishuttle systems (MSMSs). The system proposed uses overhead conveyors and active line-switching devices to transfer lines destined for different terminals, which reduces labor costs and simplifies maintenance.

Finally, we briefly specify our database search for gathering the articles to be evaluated, with 85 papers being reviewed, focusing on 2009 to 2023. As domain-specific keywords, we utilize "sortation system" and "sortation system performance". Another set of keywords, including "warehousing", "warehouse", "cross-dock", "cross-docking", "postal service", "part-to-picker", and "baggage handling system", is utilized to identify the specific domain of application. Any combination of keywords from the first and second groups was utilized as a query in the scholarly database Scopus, particularly ScienceDirect, whose primary language is English.

## 2. Synchronization Issues with Various Part-to-Picker Systems and a Literature Review

This section analyzes synchronization issues in parts-to-picker systems and examines the current literature. It concentrates on two distinct setups: SKU bins delivered from an ASRS to ergonomic picking workstations or mobile robots that deliver inventory pods to

stationary pickers. The synchronization issues of both systems are slightly different, and a categorization technique is utilized to search the literature.

### 2.1. Synchronization Problems in Warehouses

Warehouses are essential components of any supply chain, and their main responsibilities include buffering material flow, consolidating, and processing value-added materials. The order of these systems is based on two key ideas: order picking and batch picking. According to research by White John A et al. [15], the batch picking system has a higher pick rate.

Sharp Gunter P. and Fuh-Hwa Franklin Liu [16] provided an analytical technique for setting up the network of a fixed-path, closed-loop material handling system in their research. Towlines, automated monorail systems, automated guided vehicle systems, and other types of equipment can all benefit from this method. Phase one involves determining the existence of spurs, carriers needed to satisfy load movements, and guiding path shortcuts. In phase two, simulation is utilized to validate the findings and confirm the control and operational principles. CAN-Q and AGVSim are two methods for estimating vehicle requirements, and the analysis and design of material handling systems can benefit from mixed-integer programming.

Kevin R. Gue proposed a look-ahead scheduling rule to prevent long queue delays for high-cost trailers. The supervisor scans the queue and selects the first trailer with the lowest cost instead of assigning the lowest travel cost to an open strip door. This ensures that high-cost trailers are unloaded without unreasonable delay. The efficient unloading of difficult trailers is ensured by this approach. It can reduce labor costs by 15–20% compared to a first-come, first-serve policy and by 3–30% due to travel, depending on the mix of freight and queue length [17]. In the LTL sector, terminals perform three core operations: break-bulk, inbound, and outbound operations. The average workforce consists of an operations manager, 2–4 supervisors, and 2–3 dozen employees. The supervisor's look-ahead scheduling policy affects the layout of freight terminals. Thus, the author built a parametric model of material flows and utilized a local search technique to locate a nearly ideal layout [17]. Terminal managers are able to significantly reduce labor costs by scheduling incoming packaging into strip doors using a look-ahead algorithm. The algorithm is capable of providing valuable guidance for the majority of assignments and can be easily integrated into daily operations. For terminals with similar freight combinations, it may not be as effective.

Johnson M. Eric developed a model for an accumulation sortation system that utilizes analysis (AISS). In addition, he presented an analytical model for an order sortation system in automated distribution centers that utilizes stochastic elements [18]. Two common sorting strategies were identified: fixed priority schemes and the next available rule. The model shows that the next available rule performs better than fixed priority schemes in terms of sorting time and system throughput in systems with minimal lane blocking. Manufacturers are under increasing pressure to deliver reliable products, with customer satisfaction being the only thing that matters.

Centralized distribution centers serve specific markets and provide a single point of shipment for customer orders. In a distribution center, there are two main activities: incoming material storage and order picking, with order picking often being the most labor-intensive activity and cost. Many systems employ a wave approach, where a group of orders are picked simultaneously to balance the workload of shipping lanes. The model proposed in this paper examines the impact of sorting strategies on the time it takes to sort a wave of orders. According to this result, a next available sorting rule outperforms fixed priority rules in terms of wave sorting time and throughput for systems with little lane blocking. Wave generation algorithms and wave release strategies are crucial for improving system performance.

Le Anh Tuan and De Koster found that reassigning moving vehicles has a significant positive impact on lowering the average load waiting time [19]. Warehouses and manufac-



turing facilities frequently use online vehicle dispatching rules to control the movements of vehicles. The vehicle reassignment dispatching rules are effective in reducing the typical load waiting time, but frequent reassignment is not a good strategy for guided vehicles.

GU Jinxiang, Marc Goetschalckx, and Leon F. McGinnis provided a thorough analysis of the state of the art in warehouse operation planning research and identified future researched possibilities [7]. The research results are not evenly distributed among warehouse operational issues; the SLAP (storage location assignment problem) and routing have been given more attention than zoning. The focus of research should be on the operational management of warehouse systems, where multiple processes are jointly considered and multiple objectives are taken into account at once.

In 2007, Nazzal Dima and Ahmed El-Nashar reviewed research on conveyor system models used in semiconductor manufacturing facilities [20]. They provide a comprehensive introduction to simulation-based models, general analytical models of closed-loop conveyors, and pinpoints. Specific research issues and demands are addressed in the development and management of closed-loop conveyors. Furthermore, their work identifies and describes the research needs related to the implementation of closed-loop conveyor-based automated material handling systems (AMHSs). And this research concludes that the majority of the existing literature focuses on difficult-to-generalize simulation-based studies that are application-specific in nature.

Kim Byung-In et al. developed a Hungarian algorithm-based reassignment technique for overhead hoist transport (OHT). This one outperforms current reassignment-based rules and conventional dispatching rules in terms of vehicle requirements, mean lead time, and variance of lead time. It is easy to incorporate into any AGV/OHT system and can reduce overall vehicle costs by USD 1.3 million and USD 1.0 million [21].

In 2010, Gallien Jérémie and Théophane Weber focused on the issue of coordinating workflow for the completion of small orders through order release control. A quantitative model was developed to produce prescriptive control guidelines, and a rigorous performance comparison was conducted between wave-based and waveless operations [22].

Werners, Brigitte, and Thomas Wülfing proposed a robust solution approach to improve internal transports at one of the main parcel sorting centers of Deutsche Post World Net [23]. The mixed-integer linear programming (MILP) model minimizes manual transportation effort by modifying the layout and taking into account the specific characteristics and requirements of the parcel sorting center. The robustness concept is suggested to take into account uncertainty in internal transports, and the calculated robust solution is close to every optimal-scenario objective value.

The focus of this research is on the problems of release time scheduling and hub location. By the next day, the total cargo amount is guaranteed and delivered to every potential destination. This situation must meet a certain threshold and the total routing cost must be below that threshold. Yaman Hande, Oya Ekin Karasan, and Bahar Y. Kara suggested that the assumption of fixed truck release times should be relaxed and truck departures should be synchronized [24]. The assumptions of the hub network, single allocation, and flows between all pairs in the current study have drawbacks. But the elimination of these presumptions will make future research directions more challenging and achievable. The circumstance is uncertain and dynamic.

The most significant, labor-intensive, and expensive activities for warehouses are order picking and sorting operations. Kizilaslan Recep, Demet Bayraktar, and Fahrettin Eldemir presented an innovative way to integrate these operations [25]. By calculating the ideal wave size, the trade-off between picking and sorting was resolved. The purpose of this study is to aid in the decision making process regarding the ideal wave size for order picking and order sortation operations. Despite the lack of academic research on integrated order picking and sortation systems, three different operation strategies for sorting systems have been proposed.

Automated storage/retrieval systems (ASRSs) are frequently utilized in warehouses and distribution centers, according to Boysen Nils and Konrad Stephan. The effectiveness

of these systems is due to high space utilization, low labor costs, quick retrieval times, and enhanced inventory management [26]. This work proposed a unique categorization method for describing different ASRS settings and summarizing existing and innovative outcomes, based on the well-known triple notation for machine scheduling. It was determined that further investigation was necessary. Boysen Nils, Fedtke Stefan, and Weidinger Felix analyzed the scheduling of inbound trucks at hub terminals [27] to examine the advantages of an interval scheduling strategy over randomly generated solutions [27]. The sortation performance could be improved by up to 10% without spending money on technical equipment. Fedtke Stefan and Boysen Nils investigated several closed-loop tilt tray sortation conveyor design options to increase throughput in parcel distribution facilities [28]. However, it should look at different design options, as the unique destination assignment problem (DAP) described in this work significantly increases package throughput in all layout settings.

Emde Simon and Michel Gendreau presented an exact and heuristic solution to organize efficient and timely deliveries of parts and subassemblies to final assembly [29] in Figure 1. The optimization problem of scheduling in-plant transport vehicles to feed parts to mixed-model assembly lines was investigated in this paper. The result demonstrated that simple cyclic schedules are not optimal and that decreasing stopover time can decrease inventory.

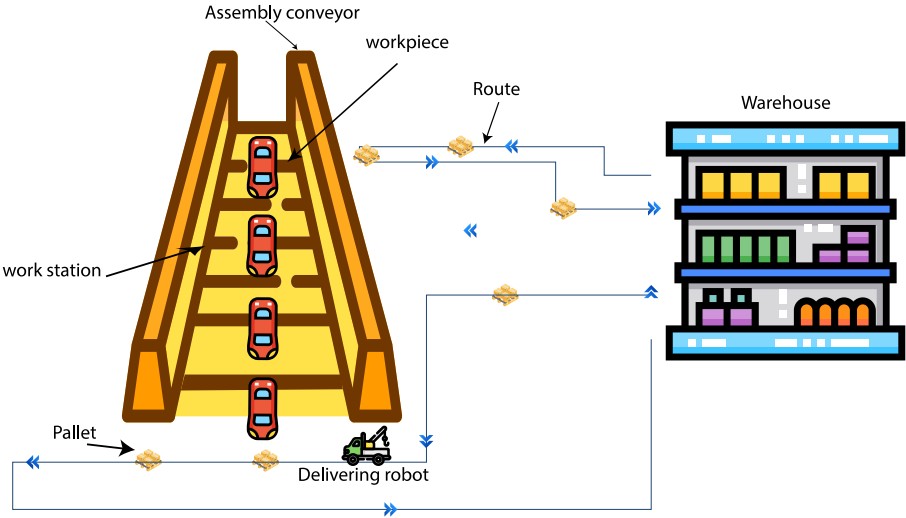

**Figure 1.** Tow trains deliver items to an assembly line [29].

Boysen Nils, Briskorn Dirk, and Emde Simon explored order processing at the picking stations in a Kiva warehouse, where mobile robots move racks straight to stationary pickers [30]. Compared to a rule-based system, efficient order processing enables one or more robots to decrease the fleet size of robots needed to promptly supply a picking station. Warehouses that store small-sized items and handle orders from a limited number of order lines are the most suitable for this system. Figure 2 describes picker-to-parts systems that minimize investment costs and are more scalable. Parts-to-picker systems eliminate wasteful picker travel and high staff expenses while minimizing the idle time of expensive equipment.

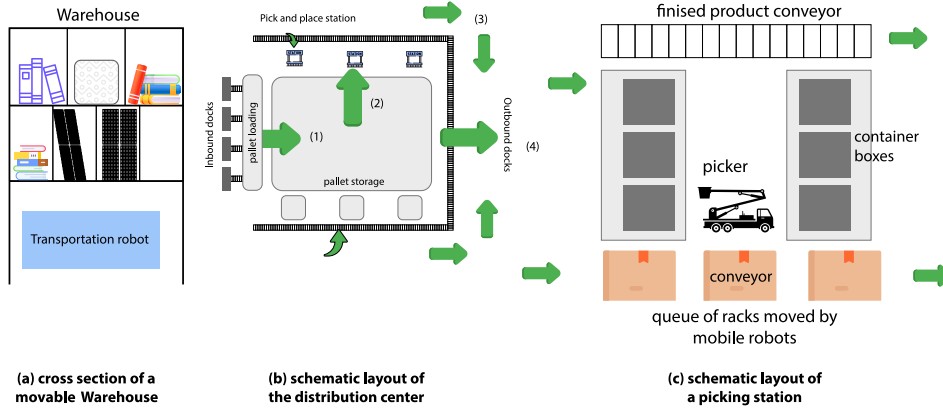

**(a) cross section of a movable Warehouse**

**(b) schematic layout of the distribution center**

**(c) schematic layout of a picking station**

**Figure 2.** The movable robot system for shifting racks [30].

Figure 3 illustrates how the order and rack sequences affect the number of rack trips needed to complete an order. Minimizing the number of rack visits is a good goal for the order picking process since it decreases the danger of bottleneck situations and the make span of the order picking schedule. Furthermore, minimizing the number of rack visits saves waiting time since all other time components are fixed once the picking list is specified and are not changed by the order sequence.

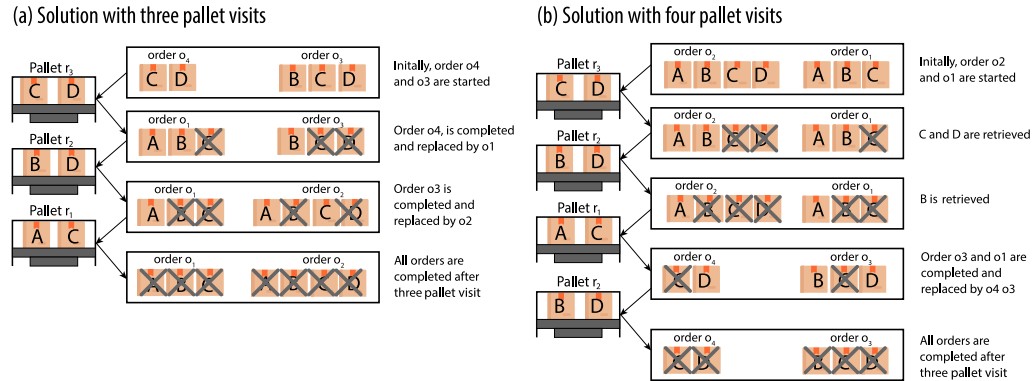

**Figure 3.** An illustration of a selection procedure based on order and rack arrangement [30].

The warehouse layout in Figure 4 has more space than is required to house m = 5 or 100 racks. But it is irrelevant for our simulation as Kiva robots can move underneath the racks. Otto Alen et al. investigated workplace design for order pickers in order to reduce ergonomic risks and improve the well-being of employees [31]. The workplace design of an order picking setting requires the integration of ergonomics aspects, which is often repeated and poses significant ergonomic risks. Computerized solution algorithms are necessary for minimizing the maximum ergonomic risk. It is important to examine ergonomic risks at workplaces, which are planned according to picking time, supplementing time measurement. The ergonomic risks of designing workplaces with acceptable physical loads should be compared to this risk.

Batching and zoning tactics are frequently used by online merchants to speed up their picker-to-parts order fulfillment processes. However, these manual consolidation processes are not scalable. Boysen Nils, Stephan Konrad, and Weidinger Felix simulated the consolidation and packing process using a put wall for manual order consolidation. It helped to reach the goal of optimizing bin release sequences and minimizing order completion durations [32]. Their objective in another study in 2018 was to evaluate a simple optimization issue to reduce the dispersion of orders in the release sequence. And this allowed for the selection of orders to be quickly assembled at packing stations [33]. To reduce order spread and evaluate the effects of various layout options, it takes into account the minimum

order spread sequencing problem. Future research should compare the configuration of the consolidation process with other configurations. The online merchant examines and separates all received products before storing them independently in a chaotic, shared storage strategy, as shown in Figure 5. Items are distributed randomly on the shelves of the picking area in a mezzanine system.

**Arrangement of the warehouse storing system**

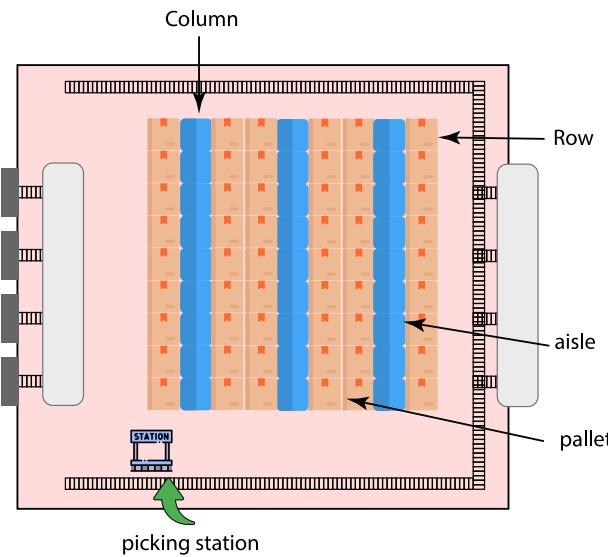

**Figure 4.** Scheme of arrangement for the pretend warehouse [30].

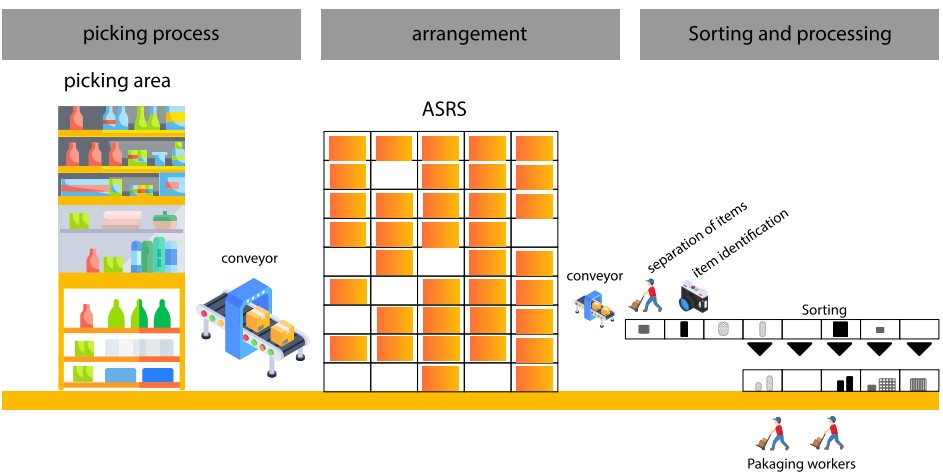

**Figure 5.** A visual representation of the warehouse industry [33].

Uriarte Claudio et al. presented various control strategies in 2019 that reduced positioning error in small-scale conveying systems. Vision-based feedback was the proposed method for controlling the right material flow on the conveyor [34]. Traditional conveyor technology is not suitable for future production and logistics systems due to its rigidity, inflexibility, less maintenance-friendliness, and cost. By examining camera depth images, the position and orientation of the objects were estimated in this paper's closed-loop control system. The object's linear and angular velocities will be selected by the controller based on the user's trajectory. The initial results are encouraging and suggest the potential of a camera-based feedback system for highly adaptable and modular conveyor systems.

New categories of robotic and automated handling systems were evaluated by Azadeh Kaveh et al, including robotic mobile fulfillment systems, shuttle-based storage and retrieval systems, and compact storage systems [8]. It concluded that new models and techniques are required to address the design and operational control challenges of such systems. And new

robotized warehouses will necessitate a review of warehouse design, planning, and control logic. Additionally, it covered the benefits of automation, including space and labor cost savings, availability around the clock, and cost savings on other operational expenses. Lastly, it focused on cutting-edge technologies like sustainable warehouses, integrated systems, automated replenishment and sequencing, and human–machine interaction.

Boysen Nils et al. examined the scientific literature on warehouse operations that met the needs of store-based retail chains in their study in 2021. Additionally, they identified the most efficient warehousing solutions for physical stores [35]. The fundamental requirements for warehouses are outlined in this document. The key decision making issues in setting up and running a unique warehousing system are clarified by checking the agreement of those requirements. Furthermore, it examined the previous literature. Pick-and-pass systems for large orders, crane-supplied pick faces, bulk picking and store consolidation, put systems, pick-to-parts, systems, and vehicle support are just a few of the topics covered. When handling high-volume-low-mix store orders and low-volume-high-mix online orders simultaneously, integrated warehouses are used to pool inventory and bundle transport. They can result in a loss of picking efficiency. Other options include establishing additional channel-specific distribution facilities, contracting with outside logistics companies, or implementing multiple parallel warehousing systems in a single distribution facility. Further study is needed to understand the order structures that suggest integrating, separating, or mixing warehousing systems.

Boysen Nils et al. classified the family of slightly varying synchronization problems in parts-to-picker systems during the following years. They demonstrated that the correct workstation setup can enhance throughput performance [14]. It is necessary to conduct research to determine whether order structures advocate integrating, separating, or combining warehousing systems when dealing with high-volume-low-mix retail orders and low-volume-high-mix online orders. The difficulty in scheduling in-plant transport trucks to supply parts for mixed-model assembly lines demonstrated that basic cyclic schedules are unsatisfactory. And plant managers should investigate more complicated scheduling processes.

Research was conducted to examine the effects of natural gas, oil, and coal energy consumption on Japan's environmental mitigation efforts and economic growth between 1965 and 2019. It was revealed that the greatest impact on environmental sustainability is caused by oil consumption. The study confirms the environmental Kuznets curve hypothesis in Japan, demonstrating that $CO_2$ emissions can predict economic growth. The analysis offers valuable policy insights for Japan's energy sector [36]. The environmental Kuznets curve (EKC) hypothesis is confirmed by the study, with a positive relationship between GDP and $CO_2$ emissions and a negative relationship between GDP2 and $CO_2$ emissions. It also shows a long-term association between $CO_2$, GDP, GDP2, and primary energy consumption in Japan [36]. Japan has made significant progress in achieving an efficient, robust, and sustainable energy system, but its carbon-intensive energy balance is still a significant obstacle. Japan is advised to diversify its energy intensity, invest in renewable energy sources, and enact policies to regulate the energy and industrial sectors, as recommended by the study. In addition, introducing clean coal technology (CCT) into coal energy systems could improve efficiency and decrease emissions.

## 2.2. Synchronization Problems in Airports

To reduce mishandled baggage, American Airlines (AA) created the largest and fastest outbound bag room at O'Hare International Airport in Chicago, as discussed in a paper by Jaroenpuntaruk et al. [37]. The stochastic computer simulation method was employed to compare four different baggage handling systems to determine which system would perform better in the future. Because the model is just a rough representation of the actual system, the simulation run may not be long enough to capture the long-term impacts of variation. The scheduling of outbound baggage at international airports is challenging due to its high complexity and resource requirements. A decomposition heuristic [38] was

suggested by Frey Markus et al. to solve this issue. The circulating conveyor belt has a capacity of a number of luggage units, and the rate of depletion is a number of operating stations per period. The decomposition of the NP-hard task of planning outbound baggage into a generalized assignment problem at international airports can lead to a solution. And it also corrects for multimedia resource-constrained project scheduling (MMRCPSP) or network flow problems.

An airport's baggage handling system (BHS) is a sophisticated system that moves, controls, screens, sorts, and stores passenger baggage from the check-in area to the departure gates. Korkmaz, Sahika Vatan, et al. used a reallocation strategy to minimize imbalances and preserve performance as a whole [39]. Different blocking rates among the input conveyors contribute to check-in system blockage issues, which diminish customer satisfaction, create airline imbalances, and degrade airport operations. To enhance capacity, it is possible to make changes to the physical design or expand the system. During the next year, Dorndorf Ulrich et al. studied a variety of mathematical models and solution techniques for solving a general flight gate scheduling problem [40]. The process is reoriented using interactive methods, while criteria aggregation is used to add new parameters.

The objective of Khosravi Abbas, Nahavandi Saeid, and Creighton Doug's research was to develop deterministic metamodels. And they can accurately and swiftly forecast the future of a complicated system such as a large baggage handling system [41]. The inspiration for it came from the trend towards complex system analysis and the incorporation of simulation modeling methods in both academia and industry. A new research area called metamodeling is addressing the shortcomings of complex and computationally expensive simulation models. To improve the estimation accuracy of metamodels, research was necessary by preclassifying data and identifying more significant factors.

Boysen Nils and Fliedner Malte introduced the aircraft landing problem (ALP). The aim was to reduce the workload of ground staff by evenly spreading the number of landed passengers, landings per airline, and landing times over the planning horizon [42]. There are two groups of ground staff working at airports: airport and airline employees. To level the workload of airport employees, a runway schedule is determined to evenly distribute the number of passengers carried by landing aircraft. Future research needs to combine leveling objectives with sequence-dependent separation times to make airport operations more efficient.

The baggage handling system (BHS) for outbound airport aircraft consists of a number of unloading zones (chutes) that are designated for departing planes. Finding ways to improve this system is necessary, such as the stochastic vector assignment problem (SVP) and the vector assignment problem (VAP). A two-stage stochastic programming technique was provided by Ilie-Zudor Elisabeth et al. to optimally solve the airport chute assignment problem, exceeding the current solution [43].

Tarău An et al. proposed an alternative approach for reducing the complexity of computations for the nonlinear optimization problem. By simplifying and approximating the mixed-integer linear programming (MILP) problem, this method was carried out [44]. This paper proposed an alternative approach for reducing the complexity of computations by approximately solving the nonlinear optimization problem using a mixed-integer linear programming problem.

A static mixed-integer model was introduced by Barth Torben and his colleagues to solve the transfer baggage problem. The proposed model combined robustness and workload allocation with efficiency goals in order to decrease missing bags and transit time in 2013 [45]. The model's operating environment includes a significant amount of problems related to uncertainties. Data analysis reveals that these uncertainties contain a significant amount of information that the model can utilize to enhance its performance. A straightforward stochastic model can be used to enhance the model, although this might be time-consuming and require changing business procedures. Simulation can be used to evaluate the value of a modification.

The airport gate assignment problem (AGAP) is a daily challenge for operations managers. Bouras Abdelghani et al. surveyed the state of the art of these problems and the various methods to obtain a solution [46]. Their work addressed the following research questions: Is AGAP NP-hard? What formulation can be defined for such a problem? The algorithms that yield an optimal solution are the exact ones. The heuristic and metaheuristic approaches use systematic rules to avoid local optimals and temporarily accept moves that cause the worsening of the objective function value. This survey examines the contributions and trends in the research using exact or approximate methods to AGAP or other related problems.

Johnstone Michael et al. analyzed how conveyor-based baggage handling systems' merging bottlenecks are designed and controlled. This demonstration shows that input variance had little bearing on bag throughput performance in their research in 2015 [47]. The effectiveness of two merging control techniques can be evaluated by measuring bag throughput. This study examined two algorithms to regulate a merge using a first-in first-out (FIFO) dispatch mechanism and found that the variable length approach outperformed the other algorithm.

In the same year, SysML was utilized to model and analyze the baggage handling system, resulting in a formal representation and validation of the system. Lin James T et al. proposed a modeling framework that uses system modeling language (SysML) to estimate the system's performance. The complex system consisted of lengthy conveyors and human decisions [48]. The integration of SysML and simulation was used to model a baggage handling system (BHS). And it also redefined critical performance for evaluation, such as maximum baggage traveling time and on-time delivery rate. Lin James T and his colleague presented an empirical study of interval release time using simulation with full factorial design of the BHS. The best combination of interval release time and its interactions was determined in this research. In addition, this also minimized congestion time in the airport baggage handling system at the Taiwan Taoyuan Airport during the 2015 simulation [49]. The simulation time for each alternative was 5.32 min, with a total time of 35.9 h required.

The outbound airport baggage handling system (BHS) consists of a set of unloading zones (chutes) assigned to outgoing flights. To enhance this system, solutions such as the vector assignment problem (VAP) and stochastic vector assignment problem (SVAP) are necessary. Huang Edward et al. provided a two-stage stochastic programming model to optimally solve the airport chute assignment problem, outperforming the current solution [50].

A hybrid heuristic was proposed by Muthukumar K. et al. to optimize the inbound baggage handling process at Munich's Franz Josef Strauss Airport [51]. It was based on an extensive simulation incorporating real-world data. And it provided a mathematical model to assign incoming flights to infeed stations, baggage carousels, and set the infeeding order of baggage tugs arriving at the same time. By doing this, the load will be balanced and passengers will receive a high level of service. This heuristic led to high-quality solutions in short running times that helped improve carousel utilization and service quality. Frey Markus, Kolisch Rainer, and Artigues Christian proposed an innovative decomposition procedure with a column generation scheme to solve practical problem instances at international airports [52]. It focused on generating a robust plan for outbound baggage handling for the entire planning day, which could be used as a basis for ground handler staffing. The proposed solution approach reduced the maximum workload obtained via the manually generated solution at the airport by 65.23%.

An airport's baggage handling system (BHS) is a sophisticated system that moves, controls, screens, sorts, and stores passenger baggage from the check-in area to the departure gates. Kim Gukhwa, Junbeom Kim, and Junjae Chae employed a reallocation strategy to minimize imbalances and maintain performance overall [53]. Different blocking rates among the input conveyors contribute to check-in system blockage issues, which diminish customer satisfaction, create airline imbalances, and degrade airport operations. To improve capacity, it is possible to make changes to the physical design or expand the system.

### 2.3. Synchronization Problems in Cross-Docking

Cross-docking is a logistics method that transfers incoming shipments directly to outgoing vehicles without storing them in between, reducing costs and consolidating shipments to achieve transportation savings. Cost reduction, shorter delivery lead time, improved customer service, faster inventory turnover, fewer overstocks, and reduced risk for loss and damage are all benefits of it.

Cross-docks require a synchronization between inbound trucks that provide products and outgoing vehicles that receive them. The synchronization process is crucial for facilitating a swift and effective trans-shipment process. Van Belle, Valckenaers, and Cattrysse [54] conducted a comprehensive survey on cross-docking, while Boysen and Fliedner [55] specialized in synchronized truck scheduling. Modeling shows that there is no discernible difference between the transportation of commodities via bins or truck trailers. As a result, the entire synchronization problem has significant similarities.

Assadi Mohammad Taghi and Mohsen Bagheri focused on identifying the optimal door assignments, docking sequences, and product assignments for the truck scheduling problem to reduce the number of late trucks at a cross-docking terminal [56]. The standard flow in a cross-docking terminal is depicted in Figure 6.

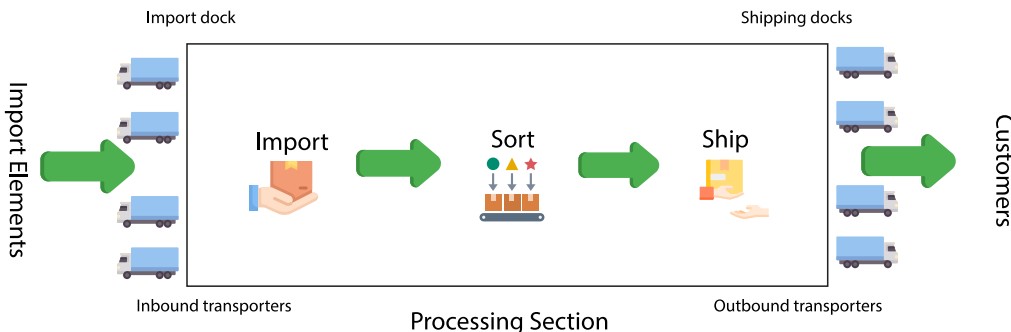

**Figure 6.** The movement of items during a typical cross-docking operation [56].

By negotiating transportation terms and reducing unit purchase costs, consolidation can reduce total logistics cost. An inventory theoretic algorithm was presented by Buffa Frank P. to evaluate the cost-effectiveness of alternative heuristics after examining the relationships between relevant variables and costs [57]. The three most important questions are what inventory grouping system is acceptable, what heuristics should be employed, and what key elements define the proper degree of consolidation.

Russell Mardi L. and Russell D. Meller provided design assistance for companies engaged in order fulfillment system design. The aim of this study was to identify the optimal door assignments, docking sequences, and product assignments for the truck scheduling problem at a cross-docking terminal. The results of this were reducing the number of late truck descriptive models based on demand levels, labor rates, order sizes, and other factors [58]. An analytical model was provided to evaluate the impact of new technology, management schemes, and worker productivity on the batching level.

The transportation of products or persons between origins and destinations is required for pickup and delivery issues (PDPs). Berbeglia Gerardo et al. provided a broad framework and a three-field categorization technique for these situations [59]. G = (V, A) represents a graph with vertex sets. V = 0, . . . , n; H = 1, . . . , p; D = (duh); K = 1, . . . , m; T V denotes trans-shipment vertices; and the total route costs were reduced. This study uses a commodity matrix to develop a framework for addressing issues such as many-to-many, one-to-many-to-one, and one-to-one. The simulation of many-to-many difficulties with trans-shipments was also possible with it. The availability of information was an essential factor in PDPs. They included static issues that are predictable and known as a priori, dynamic problems that are increasingly disclosed through time, and stochastic problems that are random variables with known distributions. See Figure 7.

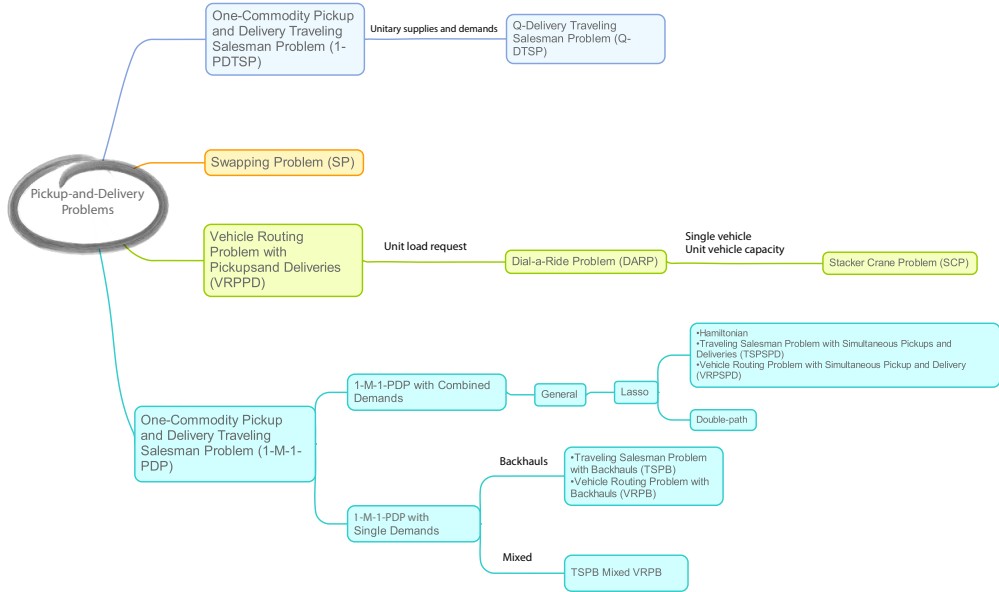

**Figure 7.** A technique for classifying pickup and delivery problems [59].

The attention on machine scheduling problems with time-dependent processing times has increased in recent years, and classical scheduling theory models have been simplified. Through research on time-dependent problems, new properties of and effective methodological approaches to algorithm design and NP-complete reduction have been discovered. A framework was presented by Cheng TC Edwin, Ding Qing, and Lin Bertrand MT to demonstrate the generalization of machine scheduling models from classical scheduling theory. And they also introduced enumerative and heuristic solution algorithms in 2004 [60]. Time-dependent scheduling is a branch of classical scheduling theory that has two common assumptions: preemptive scheduling and unit processing time scheduling.

The best truck docking or scheduling sequence for both inbound and outbound trucks was proposed by Yu Wooyeon and Egbelu Pius J. By using these, a cross-docking system [61] can reduce total operation time and maximize throughput. To minimize the total number of products that pass through the system, a heuristic algorithm was created, producing solutions that are close to the global optimal solutions, as in Figure 8.

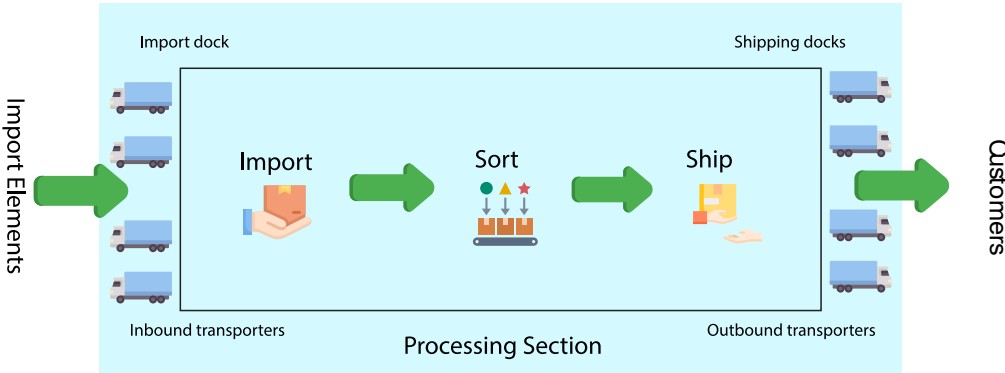

**Figure 8.** A practical operation of a cross-docking system [61].

In retail, grocery, and other distribution industries, cross-docking is a new logistics technique that transfers shipments directly from inbound to outbound trailers without any storage in between. Chen Feng and Lee Chung-Yee conducted research on a two-machine scheduling model for the logistics cross-docking problem. It was demonstrated that the problem is strongly NP-hard and that the branch-and-bound algorithm can optimally solve problems in up to 60 tasks. And the two-stage hybrid cross-docking scheduling issue

with the goal of minimizing the make span is introduced in Figure 9. Four heuristics were presented to examine the performance of moderate and large size cases. Johnson's rule-based LPT heuristic outperforms the combinational best heuristic (BH), which has an average loss of 37% in comparison to the lower limit, according to computational investigations. A technique that can be reversed to improve all heuristics and the bottom band was proposed [62].

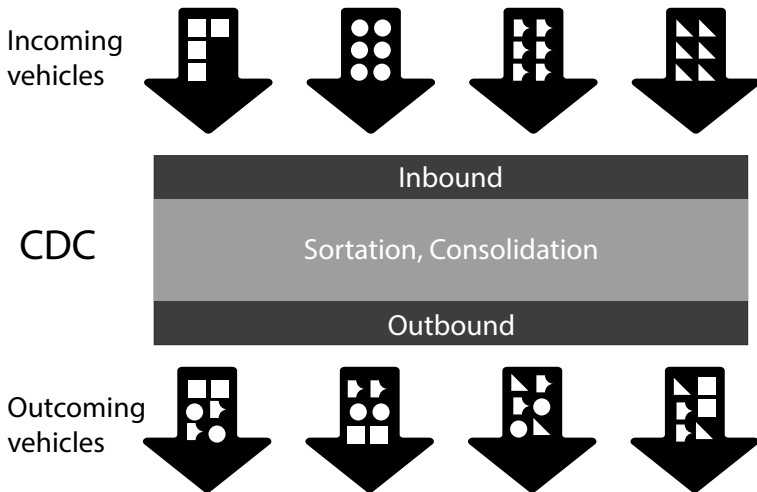

**Figure 9.** Distribution facility with cross-docks [62].

Boysen Nils and Fliedner Malte proposed a classification for deterministic truck scheduling with the objective of structuring and advancing scientific progress in real distribution networks. This method also applies to a classification of vehicle scheduling problems using tuple notation to review existing research and identify future research needs [55]. Additionally, their research in 2009 indicated that truck sequencing must be integrated into the truck sequencing problem (TRSP). Their algorithms also took into account the truck-specific processing times when loading or unloading times differ due to diverging numbers or types of carried products [63]. Cross-docks usually consist of multiple inbound and outbound doors as the Figure 10, with one side dedicated to inbound operations and the other to outbound operations. Boysen Nils developed a novel truck scheduling problem in zero-inventory cross-docking to ensure a synchronized flow of goods across the dock [64].

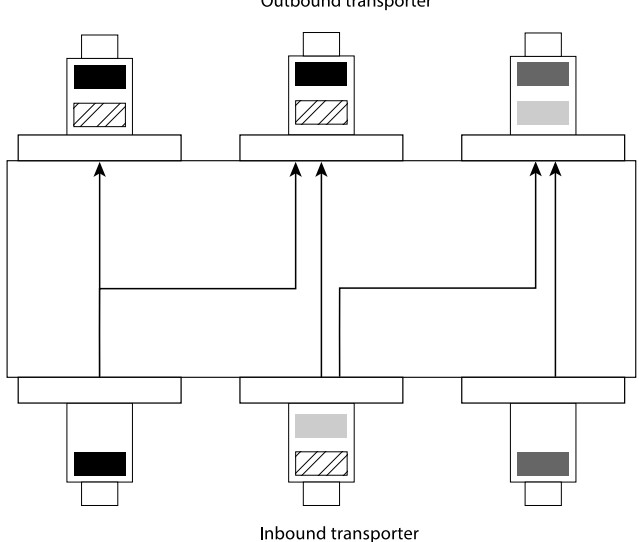

**Figure 10.** Infrastructure in cross-docking terminals [64].

Three operational objectives are considered: the minimization of flow time, processing time, and tardiness. In practice, the underlying problem setting is important because the zero-inventory policy is often used to avoid obstacles for material handling devices. Stephan Konrad and Boysen Nilssen examined the concept, categorized relevant cross-dock situations, and identified significant decision issues [65]. Terminals typically have rectangular structures with multiple dock doors, allowing incoming trucks to be loaded onto outgoing trucks and transported to their respective destinations. This activity is shown in Figure 11.

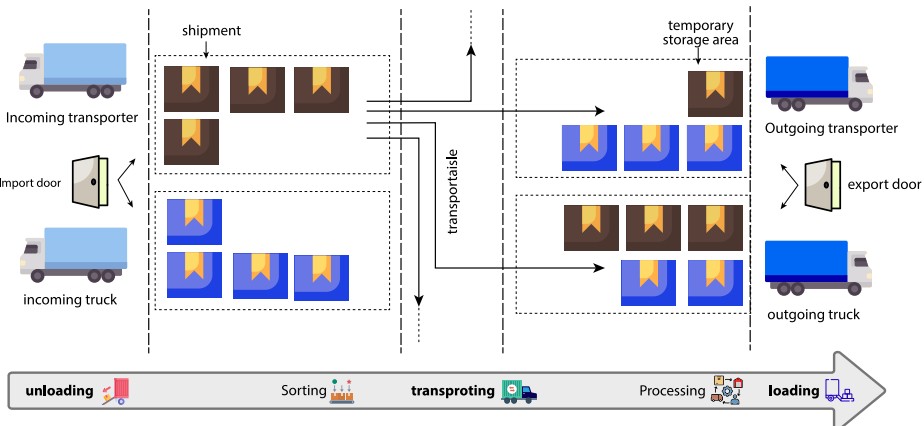

**Figure 11.** Cross-docking terminal simulation in a schematic [65].

Cargo shipments are then transferred across the dock to the temporary storage area of their respective outbound destinations. During this time, additional services are carried out as the designated outbound truck is in transit. The terminal is then loaded with trucks en route to their original destinations. Cross-docking terminals are used to unload arriving trucks, register cargo, sort it by destination, transport it across the dock, and load it onto departing trucks. Value-added services can be performed while waiting for the planned departing truck.

Shunting yards are utilized to separate freight trains and reassemble new trains, which results in time and cost savings. Boysen and his colleague also reviewed the literature on operational processes and discussed the operational challenges of freight trans-shipment [66]. The lack of investment in railway infrastructure has resulted in a gradual decrease in the percentage of freight transported via rail. A paper by Boysen Nils et al. examined container processing in railway yards from an operations research standpoint and identified unanswered research questions [67]. This paper examined the layout planning and operational decision problems that arise in rail freight yards, identifying potential research avenues for the future. The scheduling issue of inbound trucks at a cross-docking terminal was investigated by Boysen Nils, Briskorn, Dirk, and Tschöke Martin's research in 2013. Then, they developed an efficient heuristic procedure to solve it [68]. Fixed outbound schedules are essential in larger hub-and-spoke networks. To minimize delays in shipments, the paper develops two solution procedures and implements a simulated annealing procedure. Van Belle Jan et al. introduced a mixed-integer programming model that can be optimized for small-size problems [69]. To obtain good results in a short period, a tabu search approach was proposed, but uncertainty and variability were not considered. The authors advocated another approach and used the solution as an input to a logistics execution system (LES) to combine the robustness and flexibility of the LES. This resulted in the optimization of organizational objectives using the presented scheduling approach. Cross-docking is a distribution strategy that enables the consolidation of less-than-truckload shipments into full truckloads without long-term storage. By assembling full truckloads of products stored in a warehouse or distribution center, warehouses can consolidate shipments to customers. Buijs Paul and Vi's Iris FA and Carlo Héctor J presented a framework for synchronization

in cross-docking networks based on a research classification and framework which asserts that the absence of a storage buffer translates into tightly coupled local and network-wide operations [70].

A time-indexed mixed-integer linear programming formulation and a polynomial time heuristic [71] were proposed by Cota, Priscila M et al. To minimize the make span, the hybrid two-stage flow shop scheduling problem F2(P)|CD|Cmax requires identical machines and cross-docking constraints.

The just-in-time principle in warehousing involves providing the right amount of the right product at the right place at the right time, as well as synchronizing tasks. Just-in-time logistics can be subsumed by them. The just-in-time (JIT) philosophy was implemented by Assadi Mohammad Taghi and Bagheri Mohsen in truck scheduling to reduce total earliness and tardiness for outbound trucks. Their approach involved a mixed-integer programming model and two metaheuristics, namely differential evolution and population-based simulated annealing [72]. The focus was on operational issues in cross-docking, such as door assignment and truck scheduling. An overview of several scheduling techniques in this subject is provided by Józefowska [73]. Just-in-time logistics is focused on balancing just-in-time supply with expenses for small batch supply. In our case, the delivery dates are not publicly disclosed and are largely determined via the delivery order of SKU bins. The subsumption of our synchronization task into the just-in-time domain is problematic, but we do not want to discuss this philosophical issue.

Briskorn Dirk, Emde Simon, and Boysen Nils investigated the use of straightforward decision rules to determine which shipments should be loaded into which tray in automated sorting systems [74]. Fully automated sortation processes were required for many distribution networks and supply chains, which were more reliable and efficient than manual operations. They require enormous investments and are frequently bottlenecking resources, so it is essential to carefully plan the architecture and operational processes.

Rail–rail trans-shipment yards are central hub nodes that enable the rapid consolidation of containers between various freight trains, and their are four fundamental sorting systems, the alternative Megahub systems are A1, B1, A2 and B2. The A1 has 12 rail-mounted shuttle cars, B1 has a gantry crane, and B2 has a rack and shuttles with integrated lifting devices. Self-driving container storage and retrieval systems called AGVs can act as pure shuttles on rubber tires. Fedtke Stefan and Boysen Nils compared the size of shuttle fleets required by each system to deliver containers to cranes on time [75].

Colovic Aleksandra, Marinelli Mario, and Ottomanelli Michele proposed a novel network design model for green transportation by adopting a new technology called the eHighway system. To supply new overhead catenary (OC) hybrid trucks, this technology was built on electrified roads [76]. It proposes an optimization problem formulation with three objectives: the minimization of infrastructure and environmental costs and the maximization of the total number of OC hybrid trucks served on electrified arcs. The Pareto optimal approach is utilized to evaluate the Pareto front using different criteria weights and to quantify the environmental impact associated with it. Up to 99% of environmental improvement was achieved via the model by obtaining feasible and good-quality solutions.

### 2.4. Synchronization Problems in Hospitals

Methods for improving logistics performance in the healthcare industry, such as sustainable healthcare design [77] and supply chain management optimization [78], have been explored by several authors. The hospital's internal transportation has been neglected. The delivery of medicine is crucial for human health and is often based on cost objectives. Extensive research has been conducted on optimizing medicines delivery networks using multiobjective programming, mathematical models, heuristic methods, Markov processes, and metaheuristic algorithms. Medications that require temperature conditioning or decay control, such as the radioactive isotope F-18, require special attention. Lee et al. proposed a mixed-integer programming model and a large neighborhood search algorithm to reduce lead time for radioisotope delivery for cancer diagnosis and monitoring [79]. Mixed-

integer programming is a method used to model medicine delivery problems, with the aim of improving disaster response by taking into account fairness and effectiveness [80]. Erbeyoglu and Bilge's recent study proposes a novel mixed-integer linear model that aims to enhance response efforts in the event of a disaster. The model considers two crucial factors: the equitable distribution of medical supplies to affected regions and the overall effectiveness of the medication delivery process.

Chen Wanying Amanda, De Koster René BM, and Gong Yeming proposed the use of nested semiopen queuing networks (SOQNs) to model automated overhead medicine delivery in hospitals. Their attention was on the system throughput time [81].The nested queuing model is solved using two-moment approximation and an aggregation approximation method, and a simulation model is constructed to validate the analytical model under different demands. The model is accurate enough to evaluate the automated medication delivery process. And it can help the decision makers of hospitals to reduce patient waiting times and medicine response times, as shown in Figure 12. This text reviews papers on overhead hoist transport (OHT) systems used in manufacturing, particularly in semiconductor manufacturing. The focus is on the issues related to the design and operational control of AGV-based internal transportation systems. The assignment of AGVs, congestion effects, and the prevention of deadlocks are all possible. Queuing network models can capture the effects of waiting time on the system throughput time. Heuristic algorithms are being studied to solve the NP-hard problem of assigning AGVs at an automated container terminal.

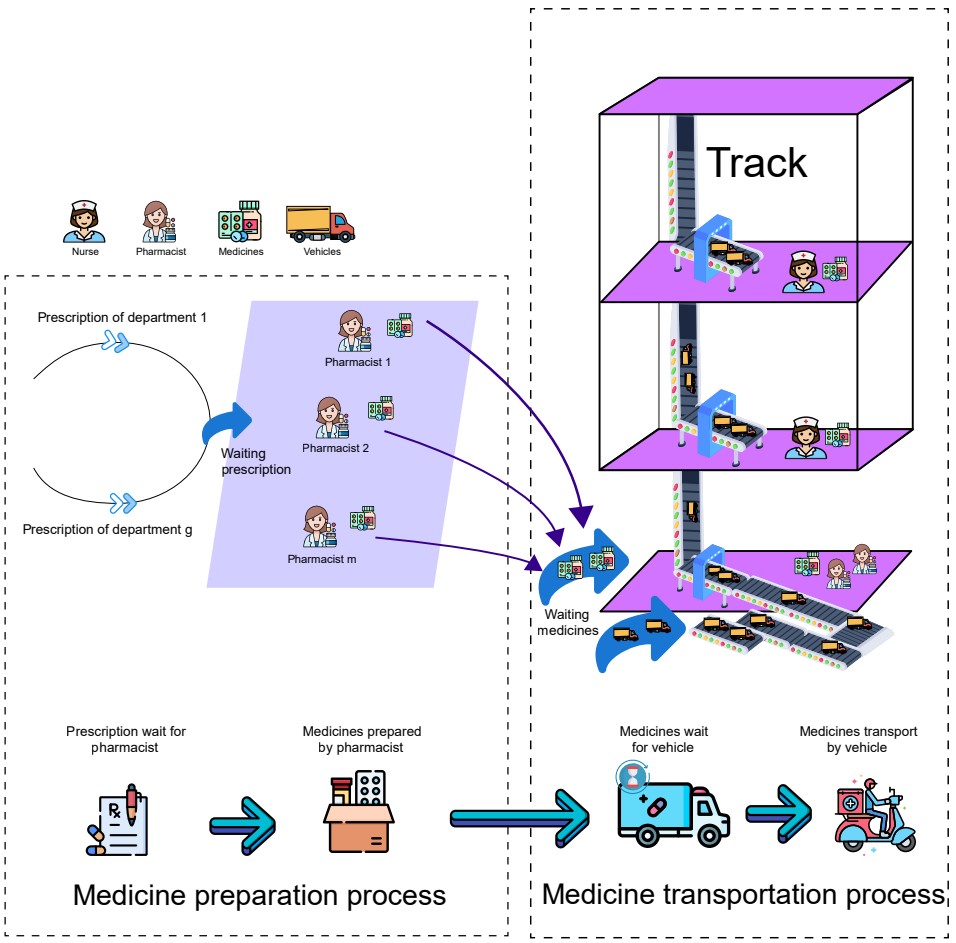

**Figure 12.** The distribution of medicine [81].

In the research of Kim, Byung-In et al., a Hungarian algorithm-based OHT reassignment technique called HABOR was proposed [21]. HABOR surpasses previous

reassignment-based rules and conventional dispatching rules in terms of the number of cars required, the mean lead time, and the variance of lead time. The overall vehicle cost can be reduced by USD 1.3 million and USD 1.0 million due to its simple execution.

Jamison M. Day examined the unique features of disaster relief efforts compared to demand-driven, steady-state supply chains [82]. The existing literature was framed using a complex adaptive supply network (CASN) lens, and eight testable propositions were proposed to improve supply network resilience. It examined the CASN-wide efficacy of deterministic supply chain management practices in disaster contexts. And it also suggested that local supply networks and dedicated disaster relief participants should investigate the collective resilience available for fostering entity properties in propositions E1–E4 and the topological hierarchy.

A biobjective min-max robust optimization model for an integrated multiperiod pharmaceutical relief network design problem was developed by Akbarpour Mina, Torabi S Ali, and Ghavamifar Ali. Furthermore, they aimed to contribute to the knowledge base on humanitarian logistics (HL) [83]. In addition to utilizing a cooperative coverage mechanism, the model takes into account the perishability of pharmaceutical goods, the mobility of relief facilities, and other factors. The predisaster multiperiod planning horizon aims to establish the best placement for central warehouses and the flow amounts of essential commodities. The analysis revealed the applicability of the model in practice, and the computational results demonstrated the effectiveness of the created robust technique.

In the paper by Khuu, N.H.L et al., the authors proposed an automatic transportation system (ATS) overhead transport model suitable for the developing country of Vietnam [84]. This ATS needs to operate efficiently in limited spaces, as well as meet the current precision machining and manufacturing capabilities in Vietnam. To guarantee automation and flexibility in product distribution, the authors proposed a multistation ATS that can redirect products to the correct stations. Automated transportation systems already exist in the world, and these systems will also be mentioned in this section. A locally produced item that is affordable and practical for upkeep in the future is necessary for this nation. This idea is being utilized by the nearby hospital as well. With the same idea with Khuu, N.H.L et al., in 2023, Truong, T.D et al. presented a novel proposition centered around mobile units capable of traversing a railway infrastructure, enabling autonomous shuttles to function individually while collaborating to accomplish various tasks [85]. The potential applications of this system span across diverse fields, including but not limited to the medical, food and beverage, automotive, and electrical appliances industries. The logistics sector faces significant competition due to the global e-commerce phenomenon, which has prompted the exploration of multistation multishuttle transportation systems (MSMSTSs). This study investigated the design of distribution networks tailored explicitly for transporting hazardous materials. The primary objective was to explore the effective integration of diverse classes of hazardous materials within these networks while also analyzing the intricate interactions between different hazardous materials. We have devised models to enhance the efficiency of logistics systems by addressing placement, routing, and inventory management challenges. These models primarily focus on solving two critical problems: the inventory routing problem (IRP) and the location inventory routing problem (LIRP). The study presented a multiperiod, two-echelon location inventory routing problem (LIRP) model that integrated time windows and vehicle fuel consumption. Subsequent investigations may direct their attention toward examining transportation systems and implementing location inventory routing problem (LIRP) methodologies about particular commodities.

Table 1 comprehensively summarizes the performance criteria and techniques differences among the studies discussed in this research. Data is organized by year and categorized into warehouses, airports, cross-dockings, and hospitals. The goal is to propose a mechanical and controller design for overhead hoist transportation systems in Vietnam to improve logistics, indoor farming, and healthcare environments.

**Table 1.** Differences between related works regarding performance criteria and methodology.

| Research Category | Article | Research Issue | Methodology |
|---|---|---|---|
| Synchronization in Warehouses | White, J.A. et al. (1989) [15] | A conveyor network equipped with divert mechanisms and accumulation lanes | Order picking and batch picking |
| | Sharp, G.P.; Liu, F.H.F. (1990) [16] | The network of a fixed-path, closed-loop material handling system | Mixed-integer programming modules of LINDO |
| | Johnson, M.E. (1997) [18] | Automated distribution center | Sort the largest (or smallest) orders first |
| | Le-Anh, T.; De Koster, M. (2005) [19] | The transportation of pallet loads at a European distribution center (EDC) | Dispatching rules (both single- and multiattribute rules) |
| | Kim, B.I. et al. (2009) [21] | The vehicle dispatching problem in large-scale overhead hoist transport (OHT) systems of semiconductor fabrication lines | Hungarian algorithm-based OHT |
| | Gallien, J.; Weber, T. (2010) [22] | The warehouses with an automated sorter from wave-based to waveless picking | Dynamic programming |
| | Werners, B.; Wülfing, T. (2010) [23] | The Deutsche Post World Net's main parcel sorting centers | Robust optimization with MILP |
| | Dijkstra, A.S. et al. (2010) [78] | Picker-to-parts warehouses | Dynamic program |
| | Yaman, H. et al. (2012) [24] | Cargo delivery at Turkey | Release time scheduling and hub location for next-day delivery problem (RSHL) |
| | Kizilaslan, R. et al. (2013) [25] | Pick-and-sort systems | The trade-off analysis between order picking and sortation operations |
| | Boysen, N. et al. (2017) [27] | Postal service industry | The optimized TSPS (truck scheduling problem in the postal service industry) and the randomized schedule |
| | Fedtke, S.; Boysen, N. (2017) [28] | Closed-loop tilt tray sortation conveyors | Greedy randomized adaptive search procedure (GRASP) and simulated annealing (SA) |
| | Emde, S.; Gendreau, M. (2017) [29] | In-house transport vehicles to feed parts to automotive assembly lines | Strong NP-completeness, and heuristic solution methods |
| | Boysen, N. et al. (2017) [30] | The order processing in a picking station | Decomposing MROP |
| | Otto, A. et al. (2017) [31] | The fast pick area of warehouses | Upper and lower bounds for ESAZ-basic, tabu search |
| | Boysen, N. et al. (2018) [33] | A conveyor system supplying the consolidation area | Minimum order spread sequencing problem with dynamic programming |
| | Boysen, N. et al. (2019) [32] | An intermediate storage system | A mixed-integer program, dynamic programming |
| | Uriarte, C. et al. (2019) [34] | Material flow operations in factories with small-scaled conveyor modules | Vision-based feedback for controlling |
| | Afshari, D. et al. (2021) [4] | Overhead crane operators | Amplitude probability distribution functions (APDFs) |
| | Bey-Temsamani et al. (2022) [2] | An intelligent overhead hoist transportation system | Follow-me principle |
| Synchronization in Airports | Jaroenpuntaruk, J.; Miller, F.G. (1995) [37] | An airport baggage facility | Simulation with common random numbers (CRNs) and antithetic variates (AVs) in GPSS/H |
| | Korkmaz, S.V. et al. (2006) [39] | An airplane cargo hold | Stowing method, tipping bags and storing; using the weight class ID |
| | Dorndorf, U. et al. (2007) [40] | Flight gate scheduling | Integer programming with objective functions |
| | Khosravi, A. et al. (2009) [41] | A baggage handling system | Artificial neural network metamodels |
| | Frey, M. et al. (2010) [38] | The outbound baggage process at international airports | A decomposition heuristic, convex cost function, NP-hard |
| | Boysen, N.; Fliedner, M. (2011) [42] | Aircraft landings | Dynamic programming approach |
| | Tarău, A. et al. (2011) [44] | Automated baggage handling systems | Mixed-integer linear programming (MILP) and the Model-predictive control (MPC) |
| | Barth, T.C. et al. (2013) [45] | Transfer baggage handling at Frankfurt Airport | Optimization under uncertainty, robust optimization, solved with a commercial MIP solver |
| | Johnstone, M. et al. (2015) [47] | Baggage handling system | The merging control algorithm: windows of fixed size moving through the merge and the size of the bag waiting to enter the merge |
| | Lin, J.T. et al. (2015) [48] | Airport baggage handling system at Taiwan Taoyuan International Airport | System modeling language |
| | Lin, J.T. et al. (2015) [49] | Airport baggage handling system at Taiwan Taoyuan International Airport | Object-oriented graphical user interface (OOGUI) simulation model |
| | Huang, E. et al. (2016) [50] | Airport baggage unloading zones | Stochastic vector assignment problem |
| | Frey, M. et al. (2017) [52] | Baggage handling at airports | Computational simulation with the Dantzig–Wolfe equation |
| | Kim, G. et al. (2017) [53] | Baggage handling at check-in area in South Korea | Computational simulation with control logic system (a reallocation algorithm) and integrated systematic view |

**Table 1.** *Cont.*

| Research Category | Article | Research Issue | Methodology |
|---|---|---|---|
| Synchronization in Cross-docking | Russell, M.L.; Meller, R.D. (2003) [58] | Automated order fulfillment systems | Modeling based on demand levels, labor rates, order sizes, and other factors |
| | Berbeglia, G. et al. (2007) [59] | Vehicle routing problems | General framework embedding a large collection |
| | Yu, W.; Egbelu, P.J. (2008) [61] | Warehouse management | Minimize total operation time or maximize the throughput of the cross-docking system |
| | Chen, F.; Lee, C.Y. (2009) [62] | Two-machine cross-docking flow shop | Polynomial approximation algorithm with an error-bound analysis |
| | Boysen, N. (2010) [63] | Inbound and outbound trucks at cross-docking terminals | Computerized scheduling and (bounded) dynamic programming |
| | Boysen, N. (2010) [64] | Zero-inventory cross-docking terminals | Dynamic programming and heuristic simulated annealing |
| | Boloori Arabani, A. et al. (2010) [9] | Cross-docking scheduling | Genetic algorithm (GA), particle swarm optimization (PSO), differential evolution (DE), and least significant difference measure |
| | Boysen, N. et al. (2013) [68] | Cross-docking terminals with fixed outbound departures | Heuristic-simulated programming (DP) and beam search (BS) for TSFD for linear mixed-integer program (TSFD-MIP) |
| | Van Belle, J. et al. (2013) [69] | Cross-docking decision support systems | Tabu search |
| | Assadi, M.T.; Bagheri, M. (2016) [56] | The truck scheduling problem in a cross-docking terminal | Mixed-integer linear programming (MILP) |
| | Cota, P.M. et al. (2016) [71] | Cross-docking center | Time-indexed mixed-integer linear programming formulation and a polynomial time heuristic |
| | Assadi, M.T.; Bagheri, M. (2016) [72] | Multiple-door cross-docking systems | Solving the Just-in-time (JIT) problem with ILOG CPLEX solver-based simulated annealing |
| | Briskorn, D. (2017) [74] | Closed-loop sortation conveyors | Numerical experiments and heuristic simulated to solve the scheduling problem |
| Synchronization in Hospitals | Chen, W.A. et al. (2021) [81] | Automated medicine delivery systems/automated overhead material handling systems | A 3D simulation model in Flexsim (of the queuing model) |
| | Khuu, N.H.L. et al. (2022) [84] | Automated medicine delivery systems/automated overhead material handling systems | Redirect products to the correct stations |
| | Truong, T.D. et al. (2023) [85] | Multistation multishuttle transportation systems/automated overhead material handling systems | The inventory routing problem (IRP) and the location inventory routing problem (LIRP) |

## 3. Discussion on the Synchronization Literature and the Multiple-Station Automatic Transportation System (ATS) for the Expectation of This Field in Vietnam

To address synchronization issues in a warehouse, it is necessary to determine if order structures favor integrating, separating, or combining warehousing systems, especially when high-volume-low-mix retail orders and low-volume-high-mix online orders are considered. The inability to schedule in-plant transport trucks to supply parts to mixed-model assembly lines demonstrates that basic cyclic schedules are not satisfactory. It is important for plant managers to investigate more complicated scheduling processes. Future study should concentrate on the following topics: (1) creating speedier heuristics and specialized accurate procedures; (2) examining ergonomic hazards in workplaces based on choosing time, combining time measurement with ergonomic risks to build workplaces with tolerable physical burdens; and (3) focusing on the operational management of warehouse systems, where diverse processes are considered jointly and several objectives are considered concurrently.

Future research should incorporate leveling aims with sequence-dependent separation times to enable efficient airport operations. A hybrid GRASP-GFLS heuristic with path relinking is proposed for inbound baggage handling at international airports to minimize carousel utilization and passenger waiting time. The problem complexity is NP-hard, and the practical application of the algorithm in a rolling planning framework is discussed.

This paper also introduced multiobjective path relinking (MOPR), a method that dispatchers can use to get solutions with different objective weights in a short computation time.

Interactive methods are utilized to realign the process, while criteria aggregation is utilized to add more factors. NP-hard is the cause of the problem. The solution can be achieved by dividing it into three problems: a generalized assignment problem, a multi-mode resource-constrained project scheduling problem (MMRCPSP), and a network flow problem with a convex cost function. More research is required to improve the accuracy of metamodel estimation, such as identifying more influential parameters and preclassifying data. Simulation can be used to access the changing value. By removing arcs and reducing graph complexity, the solution process can be sped up by developing the exact dynamic programming (DP) strategy. The simulation model will need to be more comprehensive as future studies aim to extend to other parts of this system.

The focus of cross-docking study should be on generating heuristics with better error boundaries and comparing the model to different objective functions. Preemption and mixed service modes, limited departure times, replaceable products, internal congestion, value-added activities, and robustness against deviations should be prioritized. To improve solution quality, it should focus on more advanced metaheuristics and generalizations of the truck scheduling problem with fixed outbound departures (TSFD). It is important to investigate the unifying scheduling algorithms for all four sorting systems, including pure shuttle systems and rubber-tired lift and shuttle systems. New problem solving procedures and TSFD generalizations for dealing with release dates in real-world applications should be the focus. The parcel hub scheduling problem (PHSP) should be simplified by simplifying the assumptions, such as unequal batch size trailers, linked unload and load docks, and incoming and outgoing trailer replacement periods.

A comprehensive review of multistation multishuttle systems (MSMSSs) implemented across various global locations was presented in this study. And the main focus is on their applications in the manufacturing sector. The review encompasses several key aspects. They include design considerations, operational control strategies, congestion effects analysis, deadlock prevention techniques, and queuing network models. And the heuristic algorithms employed to address the challenging NP-hard problem of assigning automated guided vehicles (AGVs) within these systems are also considered. This study examines the introduction of automated storage/retrieval systems (ASRSs). Then, the problem of storage location assignment in warehouse transportation, as well as conventional storage and order picking methods, is introduced. In addition, the mixed-integer linear programming (MILP) model has been suggested as a viable approach. Parts-to-picker systems aim to decrease picker travel and minimize the unproductive time of expensive equipment during warehouse transportation, thereby reducing both wasted picker trips and excessive personnel charges.

We also identified some overhead hoist transport models that are now utilized in hospitals, such as semiopen queuing networks (SOQNs). This can help simulate the automated overhead distribution of medicine in healthcare facilities, as well as the HABOR approach. The synchronization issues that were observed at airports were addressed by this research. And this also specifically focuses on the baggage handling system (BHS), aircraft landing problem (ALP), mixed-integer linear programming (MILP), and airport gate assignment (AGAP) difficulties. The works suggested that the vector assignment problem (VAP) and stochastic vector assignment problem (SVAP) are potential solutions for these issues. It is clear that the impact of multistation multishuttle systems on the industry and human life is significant. This research is solely focused on the analysis of a single workstation, but it is important to note that real-world systems may have several parallel workstations. Therefore, it is crucial to take into account the synchronization task when allocating orders among workstations and coordinating the demands for SKU bins across multiple workstations.

Parts-to-picker systems face uncertainty, which can include inventory discrepancies, tardy deliveries, and the effort required to locate items in mixed bins containing numerous stock-keeping units (SKUs). Understanding the influence of variables on picker search time

can lead to the development of synchronization models that are more representative of real-world scenarios. Increasing the stock-keeping unit (SKU) variety not only enhances operational flexibility but also improves the accuracy and reliability of modeling. Warehouses that utilize mobile robots to lift shelves may be able to benefit from the wide variety of racks that are available in their inventory pods.

There is a lack of literature on the subject of OHT in Vietnam. The main objective of this study was to examine the current literature on MSMSS in industrialized nations in an extensive manner. In Vietnam, hospitals frequently use suspension lines and pneumatic tube transport systems, while airports frequently use conveyor belts. It is crucial to prioritize the development of transportation systems within industrial plants and hospitals. Extensive discussion of airports is unnecessary, as their significance lies primarily in logistics. Therefore, the establishment of MSMSS in Vietnam will be guided by this evaluation as a fundamental basis. And this is particularly in utilizing the available overhead space in factories, warehouses, airports, and hospitals to mitigate transportation issues associated with material movement.

The multiple-station automatic transportation system (ATS) is the proposed solution for the expectation of this field in Vietnam based on the literature review. This ATS would ensure that nations like Vietnam, which still has a manufacturing sector with severe restrictions, can achieve it independently. The solutions for human resource development of mechanical enterprises in Vietnam were presented at the 9th International Conference on Socio-Economic and Environmental Issues in Development, NEU-KKU, 2018. The Vietnamese mechanical industry has been exposed to a number of limitations such as outdated technologies, a lack of domestic raw materials, and human resources. Insufficient management effectiveness, challenges in raising capital, developing markets, and competition also have an impact.

Managing multistation freight in-door transit is an obstacle in Vietnam, but one of the pioneering solutions to address it is to use one of the pioneering solutions. An innovative solution to the automated freight issue is provided by the proposal. A mechanical setup with overhead conveyors and active line-switching devices is used in this approach to switch lines heading to various stations. Using RFID technology that is currently available and under ideal production and manufacturing settings, this system may be utilized in Vietnam. In addition, the method ensures local manufacturing, decreases labor costs, and simplifies maintenance. The controller, the signal processing block, the information management block, the logic technique for route planning and container distribution, and other elements of this ATS design could all be thoroughly discussed in future studies.

Warehouses and hospitals in Vietnam are very old and were built decades ago, especially large institutions like Cho Ray Hospital or Hospital of Traumatology and Orthopaedics. The intensive care unit (ICU) is a hospital department that provides specialized medical care to patients who are critically ill. Since the state of disease is diverse, the number of pharmaceuticals arising during treatment is significant. In the process of dispensing and receiving drugs at the department of pharmacy, there are numerous administrative procedures that require time to prepare drugs. In particular, when a patient needs drugs during treatment, the process will follow these steps. Step 1: the doctor composes the prescription; Step 2: the nurses carry out the medical orders; Step 3: the pharmacy staff enters the prescription into the device. Then, the staff will call the warehouse to approve the slip and print the signature form with the treating doctor and the head of the department. After transferring the order to the pharmacy departments, the ICU staff give the nurse the drug back to complete the order. It is a challenging task to give medication to patients who require emergency treatment.

According to Circular 03/2018/TT-BYT published by the Ministry of Health of Vietnam on dated 9 February 2002, The Good Drug Distribution Practices require drugs to be stored and transported according to mandatory procedures. The first thing to do is to not lose information that aids in identifying the product. The product is not contaminated or adulterated by other products. Third, preventative measures are taken to prevent medicines

from being spilled, broken, embezzled, or stolen. And fourth, proper temperature and humidity conditions are maintained during transportation and storage, including using refrigeration systems for drugs that are sensitive to temperature. As a result, an automated medication delivery system is required to carry out the transportation of medicines or medical devices from one location to another in the same or separate buildings.

With these conditions, the application of transportation methods such as the use of AGVs running on the ground has great challenges due to insufficient space on the ground. The overhead space is the most suitable place to build the delivering and transporting system. The proposal of overhead suspension transport design and conductor rail transport is essential in Vietnam.

The solution is to design an overhead transport system with one single conductor rail to optimize production costs. By optimizing the path, the carriers (shuttles) can travel in different directions. Controlling the pace of each shuttle with the rail mechanism is crucial. Fuzzy logic control is a popular control strategy employed for the regulation of nonlinear systems. In addition, the sliding mode control technique is a robust and nonlinear control technique used to move an object towards a predetermined sliding mode surface within a limited duration. These algorithms can be applied in combination with remote control. The central processing station can command them wirelessly if the load of carriers (combining the weight of the car and the drug boxes) changes. Algorithms such as Dijkstra extend (each carrier will find its own shortest path) and dynamic optimization can determine the speed of each carrier and the dodge time of each carrier. And the paths of the carriers will change depending on the actual number of carriers on the table and the priority level of each carrier. This system will execute the user's commands to transport items promptly, safely, and without loss.

In emergency situations, the automated drug transport system will immediately transport drugs from the department of pharmacy and the internal medicine departments. Especially in the case of medications that require temperature-specific storage conditions, automatic transport systems will ensure quality and storage conditions due to their rapid movement. Therefore, it will enhance the efficacy of the treatment process and increase patient satisfaction.

## 4. Future Studies and Conclusions

In a previous study [84,85], we proposed another proposal for an automated transportation system (ATS). The overhead areas in warehouses, factories, and hospitals are effectively utilized by this system to conserve floor space for product transportation purposes. The purpose of this implementation is to improve operational capacity and productivity. The system consists of mechanical, control, and sensor elements. Future research should focus on optimizing the control system to enhance efficiency and reduce time consumption in the freight forwarding domain. The topic of interest is the identification of cargo and the methods used for identifying purposes. To prevent collisions during the execution of multiple tasks simultaneously, the system should employ a mechanism to divide the concurrent movement of several vehicles. It should determine the correlation between the distance traversed and the quantity of vehicles needed, as it relates to the number of stations and available shuttles. Future studies also need to focus on identifying and analyzing suitable algorithms that could effectively contribute to energy conservation. The charging system is not functioning properly. A technological framework designed to regulate and optimize the use of electrical energy stored in batteries is known as the battery power management system.

Furthermore, the Telelift system, as described, provides a high-rise conveying system that utilizes vehicles for the transportation of medications. The aim of this innovative approach is to address challenges related to staff shortages and the potential contamination of diseases. Future studies should focus on the utilization of secured, electronically monitored trucks and an interconnected network of tracks to improve the automated medication delivery process in Vietnam using queuing networks. Although the primary use of this system

is in hospitals, it can be implemented in other multilevel institutions, factories, and so on. For instance, this system can be utilized in libraries to move books or in manufacturing operations to transport various items.

Management systems, control algorithms, and algorithms for effectively controlling the simultaneous operation of various vehicles inside the system are the focus of ongoing research and development efforts. The modular mechanical design approach for stages, which includes straight movement, directional changes, and vertical transitions, has been developed to fit the specific context in Vietnam. The purpose of this system is to be installed in pre-existing spaces, with a focus on utilizing the vertical space available. The distinguishing characteristic of Vietnam's system is its ability to be deployed over various locations, necessitating the inclusion of mechanical, electrical, and control components. Furthermore, it has the unique ability to accommodate future expansion, which includes mechanical and electrical aspects, as well as control mechanisms.

The significance of bin synchronization in parts-to-picker systems is highlighted in this research. This issue has not been well addressed by the scientific material handling community or warehouse practitioners for future studies in warehouse transportation, both in manufacturing and medicine transporting. The research provides system configurations that reduce the amount of effort needed for bin supply. Furthermore, it demonstrates that synchronization based on demand could be a viable alternative to traditional supply-oriented decision tasks. In future studies, it is crucial to prioritize empirical evaluation of setup durations and comprehend the impact of bin changes and setups on picking performance. Also, the methods for resolving uncertainties related to synchronization issues. The development of novel goal functions and enhanced modeling fidelity could be a result of this, particularly in the context of e-commerce warehouses. A comprehensive study is necessary in the field of warehousing. This examines not only individual decision tasks in isolation, but also their interdependencies, particularly in the context of parts-to-picker systems. The current body of literature about layout challenges fails to adequately include the operational flexibility offered by a single SKU bin, which has the ability to fulfill the requirements of several simultaneous orders.

The performance estimates for various layout choices may be skewed or prejudiced as a result. Synchronization activities are crucial when splitting orders among workstations and coordinating SKU bin needs among many workstations. This study also provided a promised solution for Vietnam's manufacturing sectors, especially in the medical sector. Future research should prioritize the enhancement of multistation automated transportation systems (ATSs). By integrating with Vietnam's precision machining and manufacturing capabilities, these will help optimize their operational efficiency in constrained environments. The system effectively sends items to their designated stations, which facilitates automation and enhances the adaptability of product distribution processes. The item at issue, which is both cost-effective and useful, should be utilized by a nearby hospital as well as many other healthcare establishments nearby.

In addition, it is essential to incorporate an examination of emissions effects such as coal consumption, oil consumption, natural gas usage, and economic expansion on carbon dioxide ($CO_2$). The environment's sustainability is greatly influenced by transportation, which is the cause of these reasons.

**Author Contributions:** T.Q.V. is the principal author and the corresponding author. He was in charge of the main ideas to build up this research topic. He was also responsible for the overall ideas to make a literature review of the whole transportation system applying in many varied fields. He also corrected the technologies and engineering issues of this article. T.D.T. is also the principal author who wrote the first draft of this article and proposed the outline of this article. In addition, N.H.L.K. contributed to the general investigation of the application for this system in the hospital field. Q.D.L. contributed to the general investigation of the applications for this system in the airport field. H.B.T. proposed the general investigation of the applications for this system in the warehouse field. And finally, T.T.C.V. paid much attention to the real applications in the field of cross-docking. All authors have read and agreed to the published version of the manuscript.

**Funding:** This research is funded by Ho Chi Minh City University of Technology (HCMUT), VNU-HCM, under grant number B2021-20-04.

**Institutional Review Board Statement:** Not applicable.

**Informed Consent Statement:** Not applicable.

**Data Availability Statement:** The whole data-sets proposed in this manuscript are not publicly available, but they are available from the corresponding author upon reasonable request.

**Acknowledgments:** This research is funded by Ho Chi Minh City University of Technology (HCMUT), VNU-HCM under grant number B2021-20-04. We acknowledge Ho Chi Minh City University of Technology (HCMUT), VNU-HCM, for supporting this study.

**Conflicts of Interest:** The authors declare that that have no conflict of interest.

## Abbreviations

The following abbreviations are used in this manuscript:

| | |
|---|---|
| AA | American Airlines |
| AGAP | airport gate assignment problem |
| AGVs | automated guided vehicles |
| AISS | accumulation sortation system using analysis |
| ALP | aircraft landing problem |
| ANOVA | a two-way analysis of variance |
| ASRSs | automated storage/retrieval systems |
| ATS | automatic transportation system |
| AVS/R | autonomous vehicle-based storage and retrieval |
| BHS | baggage handling system |
| CASN | complex adaptive supply network |
| DARP | dial-a-ride problem |
| DP | dynamic programming |
| FIFO | first-in first-out |
| HCMUT | Ho Chi Minh City University of Technology |
| HL | humanitarian logistics |
| ICU | intensive care unit |
| IRP | inventory routing problem |
| JIT | just-in-time |
| LES | logistics execution system |
| LIRP | location inventory routing problem |
| MILP | mixed-integer linear programming |
| MMRCPSP | multimode resource-constrained project scheduling |
| MSMSTS | multistation multishuttle transportation systems |
| OBHP | outbound baggage handling problem |
| OC | overhead catenary |
| OHT | overhead hoist transport |
| PDPs | pickup and delivery issues |
| PHSP | parcel hub scheduling problem |
| RCPSP | resource-constrained project scheduling |
| RFID | radio frequency identification |
| SLAP | storage location assignment problem |
| SOQNs | semiopen queuing networks |
| SSR | shuttle-based storage and retrieval |
| SVP | stochastic vector assignment problem |
| SysML | system modeling language |
| TRSP | truck sequencing problem |
| TS | tabu search |
| TS2 | the second tabu search implementation |
| TSFD | fixed outbound departures |
| VAP | vector assignment problem |
| VNU-HCM | Vietnam National University, Ho Chi Minh City |

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
