# Peer review of "A General Overview of Overhead Multi-Station Multi-Shuttle Systems and the Innovative Applications Trend in Vietnam"

_applsci, doi:10.3390/app131911036_

Round 1

Reviewer 1 Report

The document is a descriptive narrative of a particular problem that shows up in Vietnam. Authors mention that they provide a four-folded contribution: literature review with some categorization, applicability, prospective paths and future research. It seems from the latter that such review has been missing in the vietnamese environment. I only want to recommend some numerical data because otherwise the report only enhances descriptive issues with no way to measure what authors claim. 

Thus, please include statistics or numerical data that can support what you say,

Review typos. For instance, in page 20, you wrote "siding" and it should be "sliding". Besides, authors must provide some (short) additional explanation to these types of particular concepts (fuzzy control, sliding mode control, etc).

Author Response

Dear Applied Sciences Editorial Office and Dear Reviewer,

Subject: Submission of revised manuscript ID applsci-2578822 entitled “A General Overview of the Overhead Multi-Station Multi-Shuttle Systems and The Innovative Applications Trend in Viet Nam”.

We would like to thank you very much for your email enclosing the reviewer’s comments. After carefully reading the comments, we have revised the manuscript accordingly. Below, we provide our responses in a point-by-point manner. Modifications of the revised manuscript are indicated in Italic.

Our expectation is that the revised version is now appropriate for the requirements of Applied Sciences and we are looking forward to hearing from you soon.

Best Regards,

Assoc.Prof. Tuong Quan Vo, PhD

Director, Bach Khoa Research Center for Manufacturing Engineering.

Ho Chi Minh City University of Technology (HCMUT), Viet Nam National University Ho Chi Minh City (VNU – HCM).

On behalf of all of the co-authors.

Reviewer 2 Report

First of all, I appreciate the opportunity to review the paper A General Overview of the Overhead Multi-Station Multi-Shuttle Systems and The Innovative Applications Trend in Viet Nam.  The paper deals with a very interesting problem.

My impression was, is this a review paper?!

·        The review paper must be based on a very strong and critical methodology.

·        This paper doesn’t have a methodology!

·        There is no SLR methodology. This kind of paper must have a very clear methodology (time period, journals, keywords, databases, etc). There are numerous papers for methodology (see Suggested References).

·        Everything else depends on this. 

·        The abstract is not well written. The most important results/findings must be emphasized.

·        It is necessary to understand the purpose and aim of the paper as well as its "position" in relation to previous research.

·        The separate section Practical and theoretical implications (or Discussion) is missing. The existing section Discussion is very modest. This confirms the lack of scientific and practical contributions.

·        Scientific contributions are questionable.

·        Conclusion section is not on a satisfactory level. The conclusion in scientific papers is very important.

o   Limitations of your research must be emphasized

o   Future research directions are missing.

Denyer, D. & Tranfield, D., (2009). Producing a systematic review. In D. Buchanan & A. Bryman (eds.) The sage handbook of organizational research methods. Sage Publications Inc., Thousand Oaks, CA, 671-689.

Kilibarda, M., Andrejić, M., & Popović, V. (2020). Research in logistics service quality: a systematic literature review. Transport, 35 (2), 224-235.

.

Author Response

(The authors gave the same response as above.)

Reviewer 3 Report

The manuscript entitled “A General Overview of the Overhead Multi-Station Multi-Shuttle Systems and The Innovative Applications Trend in Viet Nam”, needs extensive improvements to be accepted for publication. Kindly clarify the following queries

1. The authors need to justify the contribution of knowledge of the present work. The authors also need to validate the role of these shuttle systems from the perspective of sustainability. Kindly refer

Adebayo, T.S., Awosusi, A.A., Oladipupo, S.D., Agyekum, E.B., Jayakumar, A. and Kumar, N.M., 2021. Dominance of fossil fuels in Japan’s national energy mix and implications for environmental sustainability. International Journal of Environmental Research and Public Health18(14), p.7347.

 2.      The authors can provide some insight on AISS, specifically in the statement “Johnson M. Eric created a model for an accumulation sortation system using analysis (AISS)”. It seems incomplete

 3.      The manuscript lacks proper reference at many instances. For, e.g.Look-ahead scheduling can save 3% to 30% by cutting labor expenditures related to travel by 15-20%.”

 4.      The level of English language must be checked throughout the manuscript to make sure that the article is free from grammatical mistakes. For e.g. in the Fig-6 its “Receiving”. Fig-9 its “transporting”. Kindly enhance the quality of all the figures

5.      In the conclusion, restate your research gap, summarize the key contributions, state its significance and results, finally conclude your thoughts with the scope of any future advancements

Comprehensive English editing is required 

Author Response

(The authors gave the same response as above.)

Round 2

Reviewer 1 Report

Authors improved the document as requried. 

No problmem.

Ok.

Author Response

Dear Applied Sciences Editorial Office and Dear Reviewer,

Subject: Submission of revised manuscript ID applsci-2578822 entitled “A General Overview of the Overhead Multi-Station Multi-Shuttle Systems and The Innovative Applications Trend in Viet Nam”.

We would like to thank you very much for your email enclosing the reviewer’s comments. After carefully reading the comments, we have revised the manuscript accordingly. In the following, we provide our responses in a point-by-point manner. Modifications to the revised manuscript are indicated in Italic.

Our expectation is that the revised version is now appropriate for the requirements of Applied Sciences and we are looking forward to hearing from you soon.

Best Regards,

Assoc.Prof. Tuong Quan Vo, PhD

Director, Bach Khoa Research Center for Manufacturing Engineering.

Ho Chi Minh City University of Technology (HCMUT), Viet Nam National University Ho Chi Minh City (VNU – HCM).

On behalf of all of the co-authors.

Reviewer 2 Report

The authors made improvements. However, very confusing cover letter with 109 pages. I still can’t find the basic elements of (systematic) literature review methodology:

·        Which databases were observed?

·        Used key-words? In which part of the paper title, abstract, etc.

·        Observed period?

·        What about selection (excluding) criteria, language, etc?

I think this is important for future readers. The paper should be accepted for publication. 

.

Author Response

(The authors gave the same response as above.)

Reviewer 3 Report

Although a substantial improvements has been made in the MS "A General Overview of the Overhead Multi-Station Multi-Shuttle Systems and The Innovative Applications Trend in Viet Nam" the standard of English needs to be considerably improved.

the standard of English needs to be considerably improved.

Author Response

(The authors gave the same response as above.)
